# Walking Speed Is Better Than Hand Grip Strength as an Indicator of Early Decline in Physical Function with Age in Japanese Women Over 65: A Longitudinal Analysis of the Tanno-Sobetsu Study Using Linear Mixed-Effects Models

**DOI:** 10.3390/ijerph192315769

**Published:** 2022-11-27

**Authors:** Shunichi Ogawa, Nobuaki Himuro, Masayuki Koyama, Toshiaki Seko, Mitsuru Mori, Hirofumi Ohnishi

**Affiliations:** 1Department of Health Science, Hokkaido Chitose College of Rehabilitation, 2-10, Satomi, Chitose 066-0055, Japan; 2Department of Public Health, Sapporo Medical University School of Medicine, S-1, W-17, Chuo-ku, Sapporo 060-8556, Japan

**Keywords:** physical function, walking speed, handgrip strength, knee extension strength, older adults, aging

## Abstract

The decline in physical function with age is a major contributor to the need for long-term care. Age-related changes in hand grip strength, knee extension, and walking speed have been reported in cross-sectional studies, but longitudinal data are needed. This longitudinal study measured changes in these three measures among community-dwelling adults aged 65–89 years who participated in general health examinations between 2017 and 2019. Analyses were stratified by sex. Linear mixed-effects models adjusted for confounding factors were used to examine the interaction of different patterns of change with age of the three measures. A total of 284 participants were included in the analysis. The interaction term of age × walking speed, with age × handgrip strength as the reference, was statistically significant in women and showed different patterns in walking speed and hand grip strength. In men, none of the age × physical function interaction terms were significant in any model. For early recognition of the onset of physical function decline in older adults, any of the three measures may be used in men, but walking speed may be more suitable than hand grip strength in women. These findings may be useful in devising sex-specific screening strategies.

## 1. Introduction

Preventing a decline in physical function in older adults, especially a decline in muscle strength and walking speed, which are common determinants of the need for nursing care, is important for achieving a healthy and long-lifespan [1,2]. Hand grip strength, knee extension strength, and walking speed are good predictors of adverse health events such as disability, hospitalization, cognitive decline, and mortality [3,4,5,6,7]. Therefore, these measures are commonly used as evidence-based indicators in screening for the need for care in community-dwelling older adults [8,9,10]. In older adults, patterns of change in physical function are stronger predictors of subsequent adverse events than single observations [11]. Thus, it is important to understand patterns of age-related change in measures of physical function.

To date, research on the decline in physical function in older adults has focused on its causes and the risk of adverse events. In contrast, little is known about the patterns of change in physical function with age in older adults. Several previous studies have compared physical function in different age groups and have shown a decline in physical function with age [12,13,14,15,16]. Suetta et al. [12] reported that knee extension strength begins to decline after the age of 60 years, but hand grip strength and walking speed do not decline after age 70 years. Hayashida et al. [13] reported that knee extensor strength and walking speed decline after the age of 75 years. Taniguchi et al. [14] reported that hand grip strength and walking speed show a similar pattern of decline in adults aged 65–90 years. Most of these results are based on cross-sectional studies. However, because patterns of change with age are influenced by many confounding factors, including differences between individuals, longitudinal studies based on data of individual repeated measures are needed to assess true age-related changes. Additionally, age-related changes in physical function have been reported to vary by type of measure and to be influenced by the individual’s baseline level of physical function and biochemistry [11,12,13,14,15,16,17]. However, no reports are available on longitudinal studies that directly compare the patterns of decline between different measures of physical function in older adults after adjusting for confounding factors. Such information is important for understanding the natural course of decline in physical function in older adults, and may assist in recognizing the onset of the decline in physical function so that preventive interventions can be initiated earlier.

The purpose of this study is to compare the patterns of change in hand grip strength, knee extension strength, and walking speed with age among community-dwelling older adults using longitudinal data.

## 2. Materials and Methods

### 2.1. Procedure

The Tanno-Sobetsu study is a prospective cohort study of residents of two towns, that started in 1977. In this cohort, measurement of physical function in older participants was introduced in 2017. This analysis included residents of Sobetsu who received medical checkups in 2017 to 2019, and who were aged 65–89 years at the time of the examination. All participants who underwent measurements at any one of the three time points were included in the analysis of specific measures of physical function. The three measurements were all performed at the same time of year, at the same site, and with the same device during the health checkups. All measurements were taken under the advice of trained medical staff on the day of the checkups. Individuals who had difficulty in understanding the instructions for measuring physical function, and those who had been diagnosed with orthopedic diseases, stroke, or disorders with severe pain were excluded. Written informed consent was obtained from each participant. The study was approved by the Committee of the Institutional Review Board of Sapporo Medical University School of Medicine (24-2-21).

#### 2.1.1. Hand Grip Strength

Hand grip strength was measured with a handheld dynamometer (Takei TKK 5001, Takei Scientific Instruments Co. Ltd., Tokyo, Japan). The measurement was performed with two measurements per hand, with participants standing with their arms next to their body. The maximum value out of four attempts (in kg) was used in the analysis, with higher values reflecting better performance [18]. The weight ratio (%) was calculated as the mean of the hand grip strength on both sides, divided by weight, and multiplied by 100. The intraclass correlation coefficient(ICC) (ICC ≥ 0.88–0.92) is consistent and reported to have excellent reliability [19].

#### 2.1.2. Isometric Knee Extension Strength

Isometric knee extension strength was measured using a handheld dynamometer (Mobie MT-100; SAKAI Med, Tokyo, Japan). The participants were seated on a chair with their hip and knee joints flexed to 90°, and the force sensor was fixed to the distal side of the leg by a belt [20]. Isometric knee extension strength was measured two times in each knee. The weighted measured value in Newton-meters (Nm) was used in the analysis. The weight ratio (%) was calculated as the mean of the knee extension torque on both sides, divided by weight, and multiplied by 100. Isometric knee extension strength has excellent test–retest reliability (ICC ≥ 0.88) [21].

#### 2.1.3. Walking Speed

Walking speed, assessed by the 4 m walking test using a stopwatch, was set at an acceleration road and a deceleration road, as in a previous report [22]. Timing began and ended as the participants’ lead foot crossed the start and end points, respectively, of the 4 m walk. Participants were instructed to ambulate at their usual safe walking speed. Gait speed has excellent test–retest reliability (ICC ≥ 0.87) [22].

#### 2.1.4. Biochemical, Sociodemographic, and Anthropometric Data

Serum hemoglobin (Hb), albumin, estimated glomerular filtration rate (eGFR), triglycerides (TG), high-density lipoprotein cholesterol (HDL-C), low-density lipoprotein cholesterol (LDL-C), total cholesterol (TC), fasting plasma glucose, fasting insulin and hemoglobin A1c (HbA1c) were measured. Sociodemographic and anthropometric data were collected on age, sex, height, weight, body mass index (BMI), waist circumference, systolic blood pressure (SBP), and diastolic blood pressure (DBP).

### 2.2. Statistical Methods

The analyses were stratified by sex. After checking for normality and equal variances, Student’s *t*-test, Welch’s t-test, and the Mann–Whitney U test were used to compare each variable. Each value of the three measures of physical function were standardized by the mean and standard deviation in all participants, and then analyzed using linear mixed-effects models adjusted for confounding factors. Differences in the age-related change patterns of the three assessments were examined using interaction terms (age × measure). Four models were developed based on those used in previous studies on age-related changes in physical function in older adults, taking differences in baseline functional levels, slope of individual aging change, and biochemical data into account [17,23,24,25]. Model 1 (fixed effects only) was unadjusted; Model 2 (fixed effects + random effects (intercept)) took differences in individuals’ baseline functional levels into account; Model 3 (fixed effects + random effects (intercept + slope)) took differences in individuals’ baseline functional levels + slope into account; and Model 4 adjusted for biochemical data. The goodness of fit of the model was evaluated using the Akaike Information Criterion (AIC). When comparing the goodness of fit among nested statistical models, the lower the AIC, the better the fit of the model. Data were analyzed using IBM SPSS ver. 24 (IBM Japan, Tokyo, Japan). Statistical significance was set at *p* < 0.05.

## 3. Results

Of the 342 participants, 62 with orthopedic conditions, severe pain, or stroke were excluded. The remaining 284 participants were included in the analysis. Table 1 shows the characteristics of the study participants. Mean height, weight, BMI, waist circumference, Hb, fasting blood glucose, grip strength, and knee extension strength were significantly higher in men than in women; HDL-C and LDL-C were significantly higher in women than in men; and age, SBP, DBP, albumin, eGFR, TG, HbA1c, and comfortable walking speed did not differ significantly according to sex.

Table 2 shows the results of the model comparisons of the linear mixed-effects model in men. In Model 2, which had the lowest AIC, the estimated coefficient was higher for handgrip strength than for gait speed or knee extension strength, but the difference between the three measures was not statistically significant. The interaction terms (age × walking speed and age × knee extension strength), with age × handgrip strength as the reference were not statistically significant. The regression coefficient for age was negative and statistically significant.

Table 3 shows the results of model comparisons for the linear mixed-effects model in women. In Model 2, which had the lowest AIC, handgrip strength was significantly lower than walking speed, but there was no significant difference between knee extension strength and hand grip strength. The interaction term of age × walking speed with age × handgrip strength as the reference was statistically significant, whereas the interaction term of age × knee extension strength was not. The regression coefficient for age was statistically significant.

Figure 1 shows the age-related changes in the three measures of physical function in men and women using Model 2. Appendix A shows the age-related changes in actual measurements in the three physical function measures in men and women. Model 4, which adjusted for potential confounding factors, showed similar results to Model 2 in men and women.

## 4. Discussion

After the age of 65, the patterns of age-related changes in hand grip strength, knee extension, and walking speed were similar in men, but different patterns in walking speed and hand grip strength were observed in women. In women, the decline in walking speed started at a younger age than the decline in hand grip strength. The patterns of age-related change were similar after adjustment for other potential confounding factors. To our knowledge, this is the first longitudinal study conducted in Japan that directly compares age-related changes in these three measures of physical function in older adults.

In this study, the decrease in walking speed with age was more rapid than the decrease in hand grip strength with age in women, but not in men. In a cross-sectional study, Callisaya et al. [26] reported that the decline in walking speed with age was greater in women than in men, and the present longitudinal data support this finding. Several factors contribute to the pattern changes in walking speed observed only in women. First, women have a more significant age-related decrease in stride length and increase in the time it takes to support both feet than men, which is thought to be due to a decrease in muscle strength and the ability to compensate for balance [27]. Second, patterns of age-related change in hand grip strength are stable with a low rate of decline [15]. Since hand grip strength is a small muscle, it is thought to be less likely to undergo major numerical changes as muscle strength declines. Therefore, since women experience a more significant decline in walking speed than men, the different patterns of age-related change in walking speed and grip strength may be observed only in women.

In men, all three measures of physical function showed a similar pattern of decline with age. These results support previous studies suggesting that the pattern of age-related decline in muscle strength and walking speed are similar in older adults [16,17]. In addition to the less marked decline in walking speed with age in women than in men, several other factors may have contributed to the differences in the results between men and women. The participants in this study were from a population with a relatively high health awareness and high level of physical function, as they voluntarily participated in the health checkups. Individuals with higher baseline levels of physical function tend to experience a more gradual decline in physical function than those with lower baseline levels [16]. Thus, the participants in this study may have shown relatively small changes with age. The short observation period of 2 years.

In this study, baseline physical function levels and other potential confounding factors, including biochemical data, were considered when calculating the coefficients of age-related changes in the three measures of physical function. Suetta et al. [12] reported that the trajectory of physical function decline is determined primarily by baseline physical performance measures rather than clinical characteristics or lifestyle factors. Other studies have shown that individual differences in physical function at younger ages influence subsequent patterns of age-related change [28]. The results of this study are consistent with the results of previous studies, with Model 2, which considered an individual’s baseline level of physical function, being the best fit. In addition, Model 4 took into account other potential confounding factors including biochemical data, which may be associated with muscle strength and walking speed [24,25]. However, the results of the adjusted model were similar to those of the unadjusted model. These results suggest that baseline physical function level has a greater impact on functional decline than biochemical abnormalities. These results are also consistent with reports that general physical examination results are not associated with muscle strength or walking speed [29]. Previous studies suggest that biochemical data abnormalities are associated with decreased muscle mass, but not decreased muscle strength [23,24,25]. Muscle mass and muscle strength are not strongly correlated [14]. Moreover, diabetes in the older adult is associated with muscle weakness; thus, it is possible that diabetes accelerates age-related muscle weakness [25]. Therefore, abnormal biochemical data may be indirectly associated with reduced muscle strength and walking speed as a consequence of reduced muscle mass. Further studies are needed to examine the interaction between metabolic abnormalities and age-related decline in physical function and the long-term effects of biochemical abnormalities on age-related decline in physical function.

The results suggest that all three measures of physical function can be used for early detection of physical function decline in men aged 65 years and older, but that in women, evaluating walking speed, rather than hand grip strength, is more useful for early detection of a decline in physical function. Thus, it may be necessary to establish sex-specific strategies for selecting indicators to use in screening for the need for care in community-dwelling older adults. In addition, repeated measurement using objective measures and confirmation of changes may contribute to the early initiation of effective interventions to prevent the need for nursing care.

This study had several limitations. First, there is the possibility of selection bias. The participants underwent voluntary health checkups and may thus have had greater self-management skills and health awareness than the general population. Additionally, older adults with muscle strength below critical values may not have participated in the study due to mobility limitations. Thus, the results may have underestimated changes in physical function with age, and should be interpreted with caution. Second, the relatively short 2-year follow-up period limited the time over which changes in three measures of physical function were observed. Finally, the 284 participants in this study can be considered representative of Japanese older adults, since there were no significant differences in their characteristics compared with those of the excluded participants aged 65 and older, and hand grip strength, knee extension muscle strength, and walking speed of the participants in this study were similar to the averages of these measurements in Japanese older adults [13,14]. However, generalizability to other racial groups is limited, and further research is needed.

## 5. Conclusions

The results of this study indicate that the patterns of age-related change in hand grip strength, knee extension muscle strength, and walking speed after age 65 were similar in men, but that women showed different patterns of decline in walking speed and hand grip strength, with walking speed declining faster than hand grip strength. Because this study had a longitudinal design that took changes in individual physical function levels into account and adjusted for biochemical data, it provides an important source of reference on the patterns of age-related change patterns in physical function in older adults.

## Figures and Tables

**Figure 1 ijerph-19-15769-f001:**
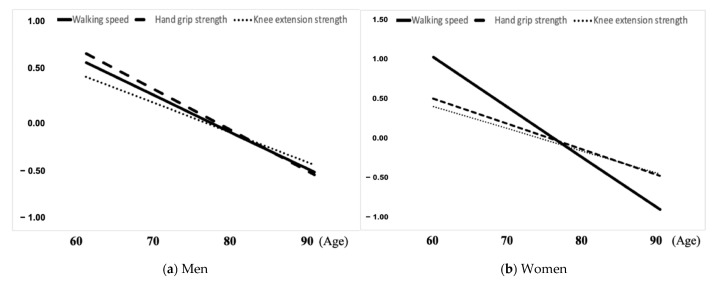
Changes in walking speed, hand grip strength, and knee extension strength with age in men and women. (**a**) Men; (**b**) Women.

**Table 1 ijerph-19-15769-t001:** Characteristics of study participants.

	Men (n = 136)	Women (n = 148)	*p*-Value
Age (years)	75.1 ± 6.1	74.7 ± 6.6	0.73
Body height (cm)	163.5 ± 6.2	149.6 ± 5.7	<0.001 *
Body weight (kg)	63.2 (41.9–108.0)	50.2 (31.9–82.7)	<0.001 *
BMI(kg/m^2^)	24.2 (15.8–32.3)	22.7(14.6–36.5)	0.013 *
Waist circumference (cm)	86.8 ± 10.5	85.7 ± 11.1	0.001 *
SBP (mmHg)	139.9 ± 17.4	141.6 ± 18.2	0.337
DBP (mmHg)	77.2 ± 9.6	75.4 ± 11.2	0.192
Biochemical data			
Hb (g/dL)	14.4 (9.9–17.6)	13.2 (8.7–15.9)	<0.001 *
Albumin (g/dL)	4.5 (3.7–5.1)	4.4 (4.0–5.2)	0.766
eGFR (ml/min)	62.9 (16.3–91.4)	60.7 (20.2–84.7)	0.381
TG (mg/dL)	92.5 (42.0–314.0)	92.5 (39.0–269.0)	0.523
HDL-C (mg/dL)	56.0(26.0–93.0)	64.0(29.0–210.0)	<0.001 *
LDL-C (mg/dL)	110.0 (49.0–171.0)	126.0 (65.0–228.0)	<0.001 *
Fasting glucose (mg/dL)	101.0 (80.0–148.0)	95.0 (80.0–152.0)	<0.001 *
HbA1c (%)	5.7 (5.0–10.5)	5.6 (4.8–8.3)	0.291
Physical function			
Handgrip strength (kg/weight)	0.65 (0.3–209.9)	0.53 (0.3–99.0)	<0.001 *
Knee extension strength (Nm/weight)	1.61 (0.6–25.5)	1.32 (0.4–17.1)	<0.001 *
Walking speed (m/s)	1.04 (0.3–1.6)	1.04 (0.2–1.8)	0.221

* *p* < 0.05. *p* values were calculated using Student’s t-test, Welch’s t-test, or the Mann–Whitney U test, as appropriate. Data are presented as the mean ± SD or median (interquartile range). Abbreviations: BMI, body mass index; DBP, diastolic blood pressure; eGFR, estimated glomerular filtration rate; Hb, hemoglobin; HbA1c, hemoglobin A1c; HDL-C, high-density lipoprotein cholesterol; LDL-C, low-density lipoprotein cholesterol; SBP, systolic blood pressure; TG, triglycerides.

**Table 2 ijerph-19-15769-t002:** Linear mixed-effects model of the association between age and measures of physical function and the interaction effects in men.

	**Model 1 (AIC = 2275.4)**				
	**Estimate**	**95%CI**	**SE**	**t**	***p*-Value**
(Intercept)	3.519	1.647 to 5.391	0.952	3.695	< 0.001 *
Age	−0.046	−0.071 to −0.022	0.125	−3.715	< 0.001 *
Evaluation (Handgrip strength)	ref				
Evaluation (Knee extension strength)	−1.571	−4.253 to 1.111	1.365	−1.151	0.250
Evaluation (Walking speed)	−0.710	−3.389 to 1.967	1.363	−0.521	0.602
Age × Evaluation (Handgrip strength)	ref				
Age × Evaluation (Knee extension strength)	0.020	−0.014 to 0.563	0.018	1.155	0.249
Age × Evaluation (Walking speed)	0.009	−0.025 to 0.044	0.018	0.525	0.600
	**Model 2 (AIC = 2230.0)**				
	**Estimate**	**95%CI**	**SE**	**t**	***p*-value**
(Intercept)	3.055	1.186 to 4.884	0.94	3.226	0.001 *
Age	−0.04	−0.064 to −0.015	0.012	−3.248	0.001 *
Evaluation (Handgrip strength)	ref				
Evaluation (Knee extension strength)	−0.892	−3.063 to 1.279	1.103	−0.808	0.420
Evaluation (Walking speed)	−0.335	−2.502 to 1.831	1.101	−0.305	0.761
Age × Evaluation (Handgrip strength)	ref				
Age × Evaluation (Knee extension strength)	0.011	−0.016 to 0.040	0.014	0.810	0.419
Age × Evaluation (Walking speed)	0.004	−0.024 to 0.033	0.145	0.303	0.762
	**Model 3 (AIC = 2249.1)**				
	**Estimate**	**95%CI**	**SE**	**t**	***p*-value**
(Intercept)	2.387	0.512 to 4.263	0.955	2.499	0.013 *
Age	−0.031	−0.056 to −0.006	0.012	−2.437	0.015 *
Evaluation (Handgrip strength)	ref				
Evaluation (Knee extension strength)	−0.868	−3.022 to 1.286	1.094	−0.793	0.428
Evaluation (Walking speed)	−0.311	−2.461 to 1.837	1.092	−0.286	0.775
Age × Evaluation (Handgrip strength)	ref				
Age × Evaluation (Knee extension strength)	0.011	−0.016 to 0.400	0.014	0.796	0.426
Age × Evaluation (Walking speed)	0.004	−0.024 to 0.032	0.014	0.286	0.775
	**Model 4 (AIC = 2274.3)**				
	**Estimate**	**95%CI**	**SE**	**t**	***p*-value**
(Intercept)	−1.024	−4.395 to 2.347	1.709	−0.599	0.550
Age	−0.031	−0.056 to −0.006	0.012	−2.455	0.015 *
Evaluation (Handgrip strength)	ref				
Evaluation (Knee extension strength)	−0.912	−3.084 to 1.259	1.103	−0.827	0.409
Evaluation (Walking speed)	−0.361	−2.528 to 1.805	1.101	−0.328	0.743
Age × Evaluation (Handgrip strength)	ref				
Age × Evaluation (Knee extension strength)	0.012	−0.016 to 0.041	0.145	0.828	0.408
Age × Evaluation (Walking speed)	0.004	−0.023 to 0.033	0.014	0.327	0.744

Model 1: Linear mixed-effects model (fixed effects), Model 2: Linear mixed-effects model (fixed effects + random effects (Intercept)), Model 3: Linear mixed-effects model (fixed effects + random effects (Intercept + Age)), Model 4: Model 2 adjusted for SBP, Hb, albumin, eGFR, TG, HDL-C, LDL-C, HbA1c. * *p* < 0.05; *p* values from the linear mixed-effects model. Abbreviations: AIC, Akaike’s Information Criterion; eGFR, estimated glomerular filtration rate; Hb, hemoglobin; HbA1c, hemoglobin A1c; HDL-C, high-density lipoprotein cholesterol; LDL-C, low-density lipoprotein cholesterol; SBP, systolic blood pressure; TG, triglycerides.

**Table 3 ijerph-19-15769-t003:** Linear mixed-effects model of the association between age and measures of physical function and the interaction effects in women.

	**Model 1 (AIC = 2554.0)**				
	**Estimate**	**95%CI**	**SE**	**t**	***p*-Value**
(Intercept)	2.592	0.807 to 4.376	0.908	2.853	0.004 *
Age	−0.034	−0.058 to −0.010	0.012	−2.861	0.004 *
Evaluation (Handgrip strength)	ref				
Evaluation (Knee extension strength)	−0.971	−3.523 to 1.580	1.299	−0.748	0.461
Evaluation (Walking speed)	1.869	−0.682 to 4.420	1.299	1.439	0.151
Age × Evaluation (Handgrip strength)	ref				
Age × Evaluation (Knee extension strength)	0.012	−0.021 to 0.046	0.017	0.738	0.461
Age × Evaluation (Walking speed)	−0.025	−0.059 to 0.008	0.017	−1.460	0.145
	**Model 2 (AIC= 2466.7)**				
	**Estimate**	**95%CI**	**SE**	**t**	***p*-value**
(Intercept)	2.434	0.687 to 4.181	0.889	2.738	0.006 *
Age	−0.032	−0.055 to −0.009	0.011	−2.738	0.006 *
Evaluation (Handgrip strength)	ref				
Evaluation (Knee extension strength)	−0.34	−2.263 to 1.582	0.977	−0.348	0.728
Evaluation (Walking speed)	2.371	0.448 to 4.294	0.977	2.425	0.016 *
Age × Evaluation (Handgrip strength)	ref				
Age × Evaluation (Knee extension strength)	0.004	−0.021 to 0.029	0.012	0.332	0.740
Age × Evaluation (Walking speed)	−0.031	−0.057 to −0.006	0.013	−2.453	0.015 *
	**Model 3 (AIC= 2477.1)**				
	**Estimate**	**95%CI**	**SE**	**t**	***p*-value**
(Intercept)	2.456	0.591 to 4.321	0.955	2.586	0.010 *
Age	−0.032	−0.057 to −0.007	0.012	−2.584	0.010 *
Evaluation (Handgrip strength)	ref				
Evaluation (Knee extension strength)	−0.413	−2.370 to 1.543	0.994	−0.416	0.678
Evaluation (Walking speed)	2.310	0.353 to 4.267	0.994	2.323	0.021 *
Age × Evaluation (Handgrip strength)	ref				
Age × Evaluation (Knee extension strength)	0.005	−0.020 to 0.031	0.013	0.400	0.69
Age × Evaluation (Walking speed)	−0.310	−0.057 to −0.005	0.013	−2.351	0.019 *
	**Model 4 (AIC= 2509.6)**				
	**Estimate**	**95%CI**	**SE**	**t**	***p*-value**
(Intercept)	0.372	−3.187 to 3.933	1.805	0.206	0.837
Age	−0.029	−0.053 to −0.006	0.012	−2.481	0.013 *
Evaluation (Handgrip strength)	ref				
Evaluation (Knee extension strength)	−0.344	−2.267 to 1.578	0.977	−0.352	0.725
Evaluation (Walking speed)	2.360	0.437 to 4.283	0.977	2.414	0.016 *
Age × Evaluation (Handgrip strength)	ref				
Age × Evaluation (Knee extension strength)	0.004	−0.021 to 0.029	0.012	0.336	0.737
Age × Evaluation (Walking speed)	−0.031	−0.057 to −0.006	0.013	−2.443	0.015 *

Model 1: Linear mixed-effects model (fixed effects), Model 2: Linear mixed-effects model (fixed effects + random effects (Intercept)), Model 3: Linear mixed-effects model (fixed effects + random effects (Intercept + Age)), Model 4: Model 2 adjusted for SBP, Hb, albumin, eGFR, TG, HDL-C, LDL-C, HbA1c. * *p* < 0.05. *p* values from linear mixed-effects model. Abbreviations: AIC, Akaike’s Information Criterion; Hb, hemoglobin; HbA1c, hemoglobin A1c; eGFR, estimated glomerular filtration rate; HDL-C, high-density lipoprotein cholesterol; LDL-C, low-density lipoprotein cholesterol; SBP, systolic blood pressure; TG, triglycerides.

## Data Availability

The data presented in this study are available on request from the corresponding author.

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
