# Peer review of "Walking Speed Is Better Than Hand Grip Strength as an Indicator of Early Decline in Physical Function with Age in Japanese Women Over 65: A Longitudinal Analysis of the Tanno-Sobetsu Study Using Linear Mixed-Effects Models"

_ijerph, 2022, doi:10.3390/ijerph192315769_

Round 1

Reviewer 1 Report

General comments: The present study addresses the physical function indicators of early decline with age which is an interesting issue.

The methods used, and the analysis performed, are adjusted to the study aim. Even though the manuscript should beneficiate from some improvements.

Introduction

It is presented satisfactorily.

Methods

In this section reliability of the used tests should be provided as also the protocol used for all the studied measurements. The reference cited contains two different methods of performing the 4m speed gait test.

 It must be stated if all measures were collected during the same day/time of the day and if any advice was provided concerning the measurement day or the previous one.

In lines 69-70 is stated “Only participants who performed measurements at any one of the three time points were 69 included in the analysis of specific measures of physical function.” Do you mean all? Please clarify.  

Concerning the measurements with the handheld dynamometer (Takei TKK 5001, 77 Takei Scientific Instruments Co. Ltd., Tokyo, Japan) please explain if the same instrument and model were used for all the measurements.

Results

In lines 135-136 add the missing verb: In Model 2, which had the lowest Akaike's Information Criterion (AIC), handgrip strength higher than walking speed and knee extension strength, but the differences be-135 tween the three measures were not statistically significant.

 Discussion

It is presented satisfactorily.

Conclusion

It is presented satisfactorily. 

Author Response

We thank you for giving us the opportunity to resubmit the manuscript. The manuscript has benefited from your suggestions, and we look forward to working with you and the reviewers to move this manuscript closer to publication in the International Journal of Environmental Research and Public Health.

Thank you for your consideration. I look forward to hearing from you.

Sincerely,

Reviewer 2 Report

The paper reports on a longitudinal study measuring age-related changes in hand grip strength, knee extension, and walking speed among community-dwelling adults aged 65–89 years who participated in general health examinations between 2017 and 2019. Analyses were stratified by sex. The results indicate that for early recognition of the onset of physical function decline in older adults, any of the three measures tested may be used in men, but walking speed may be more suitable than hand grip strength in women.

The topic is highly relevant for various disciplines and has numerous policy implications. Overall, the study is very well-designed, and the paper is well-written. There are only a few minor suggestions I would propose to the authors:

1.

“This analysis included residents of Sobetsu who received medical 67 checkups in 2017 to 2019, and who were aged 65–89 years at the time of the examination. 68”

“Of the 342 participants, 62 with orthopedic conditions, severe pain, or stroke were 120 excluded. The remaining 284 participants were included in the analysis.”

It would be helpful for the reader to mention whether the sample represents characteristics of the population surveyed (excluding those with conditions).

For example, the Discussion states:

“Finally, generalizability may be limited because this study was 233 conducted only in Japanese individuals.”

Does that mean it is generalizable to the Japanese elderly? As I understand, only respondents from two cities were surveyed. In sum, the Sample or Discussion section would benefit from an explanation of the sample and its relation to the population.

2.

“The Akaike 115 information criterion was used for model fit discrimination, with lower values indicating 116 a better fit.”

It might be useful for some readers to explain this in a sentence or two.

3.

Lines 200-217 discuss how Model 2 was a better fit than the fully adjusted Model 4. A short explanation of this result would also be useful to the reader.

In addition, based on these results, would the authors then suggest that biochemical data does not need to be measured in future studies (with these outcomes) as it has no impact on the outcome variable? (this would make future research less financially demanding).

4. While biochemical data does not impact the outcomes, is it possible that it may act as a moderator of the decline? Its potential role as a mediator was mentioned by the authors.

These are all relatively minor suggestions that the authors may want to take into account in the final version of the paper. All in all, I congratulate the authors on a well-written paper on this understudied topic using panel data, and I am looking forward to reading it in IJERPH.

Author Response

(The authors gave the same response as above.)
